# Could a Reduced Dose of 8 g of Continuous Infusion Fosfomycin Be Considered as Effective as and Safer than a Standard 16 g Dose When Combined with High-Dose Daptomycin in the Treatment of *Staphylococcal osteoarticular* Infections?

**DOI:** 10.3390/antibiotics14020139

**Published:** 2025-02-01

**Authors:** Pier Giorgio Cojutti, Sara Tedeschi, Eleonora Zamparini, Giacomo Fornaro, Manuel Zagarrigo, Massimiliano De Paolis, Pierluigi Viale, Federico Pea

**Affiliations:** 1Department of Medical and Surgical Sciences, Alma Mater Studiorum, University of Bologna, 40138 Bologna, Italy; sara.tedeschi5@unibo.it (S.T.); pierluigi.viale@unibo.it (P.V.); federico.pea@unibo.it (F.P.); 2Clinical Pharmacology Unit, Department of Integrated Infectious Risk Management, IRCCS, Azienda Ospedaliero-Universitaria di Bologna, 40138 Bologna, Italy; 3Infectious Diseases Unit, Department of Integrated Infectious Risk Management, IRCCS, Azienda Ospedaliero-Universitaria di Bologna, 40138 Bologn, Italy; eleonora.zamparini@aosp.bo.it (E.Z.); giacomo.fornaro@aosp.bo.it (G.F.); manuel.zagarrigo@studio.unibo.it (M.Z.); 4Orthopaedics and Traumatology Unit, Department of Integrated Infectious Risk Management, IRCCS, Azienda Ospedaliero-Universitaria di Bologna, 40138 Bologna, Italy; massimiliano.depaolis@aosp.bo.it

**Keywords:** fosfomycin, continuous infusion, therapeutic drug monitoring, hypokalemia, hypernatremia, PK/PD target attainment

## Abstract

**Background**: Daptomycin plus fosfomycin combination therapy is a valuable strategy for treating staphylococcal osteoarticular infections (OIs), but hypernatremia and hypokalemia due to sodium overload are important issues. The aim of this study was to assess the likelihood of attaining a pharmacokinetic/pharmacodynamic (PK/PD) target of AUC/MIC > 66.6 and/or of 70%t > MIC with continuous infusion (CI) fosfomycin at the recommended vs. reduced dose in patients with OIs receiving combination therapy with high-dose daptomycin. Adverse events were also evaluated. **Methods**: Patients with OIs treated with 8–10 mg/kg daily daptomycin plus CI fosfomycin, and who had a ≥1 TDM assessment of CI fosfomycin, were retrospectively included in the high-dose (16 g daily) or reduced-dose (<16 g daily) groups. The attainment of the PK/PD targets of 70%t > MIC and AUC/MIC > 66.6 up to an MIC of 32 mg/L was calculated. A CART analysis was used to identify a cut-off of fosfomycin AUC that indicated occurrence of hypernatremia and/or hypokalemia. **Results**: A total of 44 and 39 patients were included in the high- and reduced-dose groups, respectively. The two groups did not differ in terms of demographic characteristics, underlying infectious diseases and microbiological isolates. No differences between groups in attaining both PK/PD targets up to an MIC of 32 mg/L and in C-reactive protein reduction at the end of treatment were observed. Fosfomycin AUC > 8245 mg × h/L and >8326 mg × h/L were associated with hypernatremia and hypokalemia, respectively. **Conclusions**: CI fosfomycin at 8 g daily may reach optimal PK/PD target attainment with better safety than the recommended 16 g daily dose in patients with preserved renal function. Targeting fosfomycin AUC at 2131–8326 mg × h/L or steady-state concentration at 88.8–347 mg/L may be adequate for optimizing drug pharmacodynamics up to an MIC of 32 mg/L and minimizing the risk of hypernatremia and hypokalemia.

## 1. Introduction

Osteoarticular infections (OIs) represent one of the most difficult-to-treat bacterial infections, associated with chronic disability and high healthcare costs. Their burden is increasing due to aging populations, increasing incidence of chronic comorbidities and widespread use of arthroplasty surgery. OIs include a heterogeneous group of diseases, consisting of infections of the native tissues (such as vertebral osteomyelitis, septic arthritis, diabetic foot infections and acute hematogenous osteomyelitis) together with implant-associated infections (such as prosthetic joint infections, fracture-related infections and spinal implant infections). Pathogenesis varies according to infection type and the main mechanism of infections, this being spread from contiguous surrounding tissue, direct inoculation of microorganisms during surgery or traumatic injury, or hematogenous spread from bloodstream infections [1,2].

Many microorganisms are involved in OIs. Overall, *Staphylococcus aureus* and coagulase-negative staphylococci (CoNS) may cause up to two-thirds of all infections, with *Staphylococcus aureus* being the most prevalent single pathogen. Among them, methicillin-resistance has been reported to occur in as high as 50% of strains [3]. Less frequent pathogens may include streptococci, enterococci, Gram-negative bacilli and anaerobes. Polymicrobial infections may be possible and may occur more frequently in cases of diabetic foot infections (up to 80% of cases) and of post-traumatic infections (approximately 30%) [4].

Intravenous (i.v.) fosfomycin is regarded as a valuable option for the treatment of bone and joint infections because of its good bone penetration rate and its wide spectrum of activity, including both methicillin-resistant *Staphylococcus aureus* (MRSA) and Gram-negative bacteria [5,6]. Experimental studies have showed that fosfomycin has a valuable synergic effect with daptomycin against staphylococci [7,8] and a considerable anti-biofilm activity [8,9]. Consequently, the combination of high-dose daptomycin plus fosfomycin is nowadays considered a valuable approach for starting treatment of staphylococcal OIs and abating the bacterial burden in the first weeks of treatment [10].

Which pharmacokinetic/pharmacodynamic (PK/PD) determinant of fosfomycin may correlate best with microbiological eradication and clinical efficacy against staphylococci is still a matter of debate. Some authors found that maintaining concentrations above the MIC for more than 70% of the dosing interval (70%t > MIC) was associated with good antibacterial activity [11,12,13]. Others showed that an area under the concentration–time curve to MIC ratio (AUC/MIC) > 66.6 was associated with a 2-log drop of staphylococcal bacterial load at 24 h in an in vitro model of infection [14,15].

Fosfomycin is licensed for the treatment of OIs at a standard daily dose of 12–24 g in patients with normal renal function, and dosage reduction is recommended only in patients having renal dysfunction with an estimated glomerular filtration rate (eGFR) < 50 mL/min [16]. However, considering that i.v. fosfomycin has a disodium formulation, excessive sodium load, induced by causing sodium overload and electrolytic imbalance, could be an issue, especially in patients with heart failure and/or in the comorbid elderly. In this regard, it should not be overlooked that, with 330 mg being the sodium content per each gram of disodium fosfomycin, the use of fosfomycin dosages in the upper part of the licensed posology, namely 16–24 g, could pose some safety issues.

Interestingly, in a recent population pharmacokinetic study, we showed by means of Monte Carlo simulations that using a reduced daily dose of 8 g daily administered by continuous infusion (CI) may be sufficient for attaining an optimal PK/PD target of 70%t > MIC against *Staphylococcus aureus*, including MRSA, in the vast majority of patients with OIs. Consequently, we started using reduced daily dosages of 8–12 g by CI and measuring concentrations by means of therapeutic drug monitoring (TDM) for testing whether this approach may be valuable in granting similar PK/PD target attainment and better safety compared to higher standard dosages in the management of OIs [17].

The aim of this study was to assess the likelihood of attaining a pharmacokinetic/pharmacodynamic (PK/PD) target of AUC/MIC > 66.6 and/or of 70%t > MIC with continuous infusion (CI) fosfomycin at the recommended vs. reduced dose in patients with OIs receiving combination therapy with high-dose daptomycin. Adverse events were also evaluated.

## 2. Results

### 2.1. Patient Population

A total of 83 patients were included in this study; 53.1% (44/83) in the high-dose group) and 46.9% (39/83) in the reduced-dose group. Their demographic and clinical characteristics are reported in Table 1.

The median (min–max) age and BMI of patients in the high-dose group were 56 (18–81) years and 27.6 (17.9–44.5) kg/m^2^, respectively. In the reduced-dose group, the median (min-max) age and BMI were of 59 (18–82) years and 26.4 (14.5–38.1) kg/m^2^, respectively. The two groups did not differ in terms of demographic characteristics, underlying infectious diseases and microbiological isolates. Baseline laboratory parameters were also similar, apart from renal function. The estimated glomerular filtration rate slightly differed between the two groups, but all patients had a preserved renal function [median (min-max) eGFR of 98.5 (32–126) and 108 (67–134) mL/min/1.73 m^2^ in the reduced- and high-dose group, respectively].

Prosthetic joint infections and osteomyelitis were the two most prevalent infections, accounting overall for 65.9% (29/44) and 69.2% (27/39) in the high- and reduced-dose groups, respectively. Microbiological isolates were identified in 50% (22/44) and 33.4% (13/39) of patients in the high- and reduced-dose groups, respectively. The most frequent clinical isolates were *Staphylococcus aureus* [29.5% (13/44) of cases in the high-dose group] and coagulase-negative staphylococci [23.1% (9/39) of cases in the reduced-dose group].

Median daily (min–max) fosfomycin doses were 16 g (16–16 g) in the high-dose group and 8 g (8–12 g) in the reduced-dose group. In the reduced-dose group, compared to the high-dose group, the median fosfomycin clearance was higher (2.3 vs. 1.8 L/h, *p* = 008, respectively) and the median steady-state concentration (Css) was lower (157.8 vs. 381.7 mg/L, *p* < 0.001, respectively). Figure 1 shows the distribution of fosfomycin Css in each group. A high inter-patient variability was observed in fosfomycin Css, with the CV% being of 70 and 110% in the high- and reduced-dose group, respectively.

In patients with eGFR > 70 mL/min/1.73 m^2^, patient clearances had a 12.6-fold variation, ranging from 0.35 to 4.44 L/h.

### 2.2. Pharmacokinetic/Pharmacodynamic Analysis

The percentage of desired PK/PD target attainment of fosfomycin in the two groups, both in terms of 70%t > MIC and of AUC/MIC > 66.6, in relation to the EUCAST MIC_50_ and MIC_90_ of *Staphylococcus aureus* and to an MIC value of 8 and 32 mg/L, are summarized in Table 2. In the high-dose group, all patients attained both of these targets, whereas, in the reduced-dose group, all patients attained a 70%t > MIC and a 92.3% AUC/MIC > 66.6.

The distribution of C-reactive protein (C-RP) at baseline and at the end of fosfomycin treatment is depicted in Figure 2A and Figure 2B for the high- and reduced-dose group, respectively. No significant differences were observed in the median C-RP value at baseline in the high- vs. reduced-group (9.66 vs. 10.88, *p* = 0.392) as well as in the median C-RP value at the end of treatment in the high- vs. reduced-dose group (2.0 vs. 2.75, *p* = 0.644).

Clinical evaluation of antimicrobial treatment at time of switching was similar in the two groups, with the treatment efficacy being 90.9% (40/44) in the high-dose group and 89.7% (35/39) in the reduced-dose group.

### 2.3. Assessment of the Safety Profile

The safety profile of antimicrobial treatment in the two groups is reported in Table 3. Hypernatremia was observed at a higher proportion in patients in the high-dose compared to the reduced-dose group, but the difference did not reach a statistical significance (11.4 vs. 2.6%, *p* = 0.207). Interestingly, hypokalemia trended to be prevalent in the high-dose group compared to the reduced-dose group (15.9% vs. 2.6%, *p* = 0.06). Overall, fosfomycin exposure in terms of AUC was higher in patients experiencing electrolyte imbalances compared to those who did not. Specifically, in patients with vs. without hypernatremia fosfomycin AUC was 10,942 vs. 6648 mg × h/L (*p* = 0.037), respectively, while in patients with vs. without hypokalemia it was 9744 vs. 6640 mg × h/L (*p* = 0.015), respectively.

Interestingly, a classification and regression (CART) analysis allowed for identifying cut-off values of fosfomycin AUC as valuable predictors for both hypernatremia (AUC > 8245 mg × h/L, Figure 3) and hypokaliemia (AUC > 8326 mg × h/L, Figure 4) occurrence.

## 3. Discussion

This is the first real-life study comparatively assessing the PK/PD target attainment and safety profile of high-dose vs. reduced-dose continuous infusion fosfomycin combined with daptomycin in a cohort of patients with osteoarticular infections. Our findings suggest that reduced dosages of fosfomycin may be sufficient for reaching optimal PK/PD targets against staphylococci with an MIC up to 32 mg/L while, at the same time, achieving a similar C-RP reduction at the end of treatment and minimizing the occurrence of adverse events.

Different randomized clinical trials and observational studies evaluated the clinical role of fosfomycin in different clinical scenarios [18,19,20,21,22]. However, in the setting of bone and joint infections most studies confirmed the efficacy of combination therapy with fosfomycin [21,22,23].

Despite being discovered in 1969 and its consequent long life in the market, there is still scarce information on most essential aspects of fosfomycin pharmacodynamics. First, for patients with preserved renal function, there is great variability in the recommended dose amount, namely from 12 to 24 g daily, and mode of administration [16]. This latter aspect has recently been the aim of two prospective population pharmacokinetic studies that showed the superiority of CI in reaching the 70%t > MIC PK/PD target [17,24]. Monte Carlo simulations showed that doses of 8–12 g may be sufficient against MSSA and MRSA infections, assuming EUCAST MIC distribution for the cumulative fraction of response calculation [17]. The second aspect is related to which PK/PD target should be considered. Some authors have suggested that fosfomycin behaves like a time-dependent antibiotic, especially against *Enterobacterales*, and proposed a 70%t > MIC target, albeit with low supporting evidence [11,12,13,14,25,26]. More recently, Noel et al. observed, in an animal model of staphylococcal infection, that an AUC/MIC > 66.6 was the best PK/PD parameter associated with a 2-log reduction at 24 h in a viable count of *S. aureus* [15]. Therefore, it appears reasonable to consider this parameter when optimizing fosfomycin dosing in the context of staphylococcal infections. The third critical issue is related to the unavailability of pathogen MIC even in most tertiary care hospitals, as fosfomycin microbiological testing requires the agar dilution method [27], and this hampers the estimation of patient-specific AUC/MIC. Finally, it is worth noting that, for fosfomycin, EUCAST removed the clinical breakpoint, which was formerly set at 32 mg/L. For this reason, we arbitrarily considered the MIC_50_ and the MIC_90_ for PK/PD calculation in our population even if a cut-off value of 32 mg/L may be still considered to be effective against less susceptible strains. However, when looking at the EUCAST MIC distribution, only 5% (30/604) of staphylococcal strains have MIC > 16 mg/L, namely the MIC_90_.

In this regard, our study may also contribute to shedding some light on which is the lower concentration of fosfomycin that should be a target for TDM purposes. Indeed, to target an AUC/MIC > 66.6 at an MIC of 32 mg/L, an AUC of 2131.2 mg × h/L, which corresponds to a Css of 88.8 mg/L, should be achieved in plasma. Overall, few studies have reported on the use of low dosages of fosfomycin in the treatment of OIs. The possibility of using lower-than-standard doses in this setting was proposed first by Rodriguez-Gascon et al., who suggested using a loading dose followed by maintenance doses of <16 g daily by continuous perfusion [28]. Subsequently, Luengo et al. described a successful treatment over 42 days with 8 g daily fosfomycin combined with 10 mg/kg daily daptomycin in an elderly patient with a difficult-to-treat infection of a total femoral replacement caused by multi-drug-resistant *Staphylococcus epidermidis* [29].

Most studies are aligned in observing that hypernatremia and hypokalemia are the most frequent adverse event of i.v. fosfomycin. A French study that included 72 patients treated with i.v. fosfomycin mostly for bone and joint infections with a median dose of 12 g daily for a median duration of 11 days showed an overall prevalence rate of adverse events of 38% and of hypokalemia of 26% [30]. Rates of hypokalemia ranging from 4.3 to 28.57% and of hypernatremia ranging from 10.5–41.9% [31,32,33,34] were also reported by studies including different patient populations. Our findings of hypokalemia and hypernatremia being at 9.6% and 7.2%, respectively, are consistent with these data.

While hypernatremia may be easily explained by the extra sodium intake related to fosfomycin, hypokalemia is believed to be due to an increase in urinary potassium excretion within the distal tubule [6]. Electrolyte imbalance may be a serious issue, especially in patients with pre-existing heart diseases or renal failure. However, in most cases, both hypernatremia and hypokalemia are mild and lead to discontinuation in less than 20% of cases [31]. Similarly, the role of potassium supplementation while receiving fosfomycin therapy remains unclear [32]. Nevertheless, in some recent randomized trials evaluating the role of combination fosfomycin in the treatment of MRSA bacteremia and endocarditis [19] and the effectiveness of fosfomycin in the treatment of *Escherichia. coli* bacteremic urinary tract infections [35], significantly more adverse events leading to treatment discontinuation occurred in the fosfomycin arm, including heart failure.

Biscarini et al., in their study of a mixed population of 224 patients with different infections, underlying diseases and microbiological isolates, showed that the occurrence of adverse events due to fosfomycin was associated with ICU admission, the presence of deep-seated infections and septic shock in multinomial regression analyses [31]. Among the 68/224 patients who had at least one TDM assessment of fosfomycin concentration, they also showed that patients who developed hypernatremia had median trough concentration (Cmin) values that were significantly higher than those who did not (419.5 vs. 140.6 mg/L, *p* = 0.012), and a trend of higher median values was also noted for Css (294 vs. 146 mg/L, *p* = 0.103, respectively) [31]. Our CART model confirmed this observation for hypernatremia and extended it also for hypokalemia with good reliability and consistency. In particular, the AUC cut-off value of 8245 mg × h/L for hypernatriemia, corresponding to a Css of 343.5 mg/L, and the AUC of 8326 mg × h/L for hypokalemia, corresponding to a Css of 346.9 mg/L, may be regarded as the plasma therapeutic threshold for toxicity when adjusting fosfomycin doses through TDM.

TDM of fosfomycin has been recently advocated as a useful tool in order to optimize fosfomycin exposure, but few clinical experiences have been reported so far. Recently, our group described a patient with ventilator-associated pneumonia and bacteremia caused by KPC/OXA48-producing *Klebsiella pneumoniae* who was successfully treated with CI fosfomycin and meropenem by means of TDM-based dose adjustments of both antibiotics. In particular, the fosfomycin dose was reduced from 24 g daily to 16 g daily, and microbiological eradication was conformed at day 7 and 15 days [36]. We also described a case of another patient treated with CI fosfomycin and ceftazidime/avibactam for post-neurosurgical ventriculitis caused by carbapenem-resistant *Klebsiella pneumoniae* who was successfully treated by TDM-based dose optimization in order to target the PK/PD parameters of efficacy both in plasma and in the CSF [37]. Successful treatment outcomes in six patients treated with continuous infusion fosfomycin and cefiderocol optimized by TDM for difficult-to-treat *Pseudomonas aeruginosa* infections were also reported by us. In that cohort, the median fosfomycn Css was 504.9 (363.2–647.2) mg/L, but, unfortunately, no fosfomycn-related adverse events were retrieved [38]. An observational German study on 17 patients with ventriculitis treated with 24 g continuous infusion fosfomycin combined with meropenem and vancomycin showed median plasma Css and AUC values of 209 (163–438) mg/L and 4800 (3816–7152) mg × h/L, respectively [39]. Interestingly, the mean (±SD) fosfomcyin clearance in this population was of 4.2 ± 2.2 L/h, which is higher than in our patients despite similar renal functions. Although fosfomycin is eliminated mostly by the renal route, the relationship between fosfomycin clearance and eGFR is not linear but exponential [39], and a high variability in the plasma concentrations has been observed both in this and in our cohort. Even if this aspect deserves further investigation, the unpredictability of plasma concentrations in patients with preserved renal functions may require TDM for reducing the risk of fosfomycin toxicity.

We acknowledge some important limitations of this study. First, the retrospective design and the presence of possible confounders should be recognized. Specifically, we did not retrieve patients’ co-treatments with other drugs so that the eventuality that other agents could have concurred to cause hypokalemia, namely corticosteroids or tiazidic diuretics, could not be ruled out [40]. Second, the small sample size may have limited the statistical power for detecting the difference in the proportion of drug-related adverse events between the two groups. However, the association between plasma exposure and both hypernatremia and hypokalemia occurrence was significant and represents a new finding with clinical implications. The unavailability of fosfomycin MIC may have precluded the determination of patient specific PK/PD targets for a more personalized dose approach. Finally, we recognize that prospective confirmatory studies are needed, especially in relation to the identified threshold of toxicity.

## 4. Materials and Methods

### 4.1. Study Design and Clinical Setting

This retrospective, observational, monocentric study was carried out in patients admitted to the IRCCS, Azienda Ospedaliero-Universitaria di Bologna, Bologna, Italy, between 2020 and 2024 who had OIs treated with daptomycin plus CI fosfomycin and who were undergoing TDM of fosfomycin.

At our center, a multidisciplinary team is specifically dedicated to optimizing the management of OIs in both inpatient and outpatient clinics. This team is coordinated by infectious disease consultants and involves either orthopedic surgeons for interventions or MD clinical pharmacologists for TDM-based antimicrobial treatment optimizations.

Usually, the management of OIs depends on several factors, and, for an extensive description of these, the readers are referred to a previously published study [10]. Briefly, antibiotic therapy is started only after collecting samples from the infection site for performing microbiological cultures (bioptic sampling in cases of vertebral osteomyelitis with no surgical indication or intra-operative sampling in the other types of infections). Afterwards, empirical intravenous antibiotic therapy is always started with anti MRSA/MRSE empirical treatment based on daptomycin plus fosfomycin. If additional risk factors for ESBL-producing Gram-negative bacteria exist, ertapenem is also added. Empirical treatment is maintained and remains unchanged until feed-back from the culture results is obtained. Whenever cultures are positive for a given pathogen, therapy is promptly switched to target treatment (by the oral route whenever feasible); otherwise, only anti MRSA/MRSE coverage with i.v. long-acting antibiotics (i.e., dalbavancin) or with oral agents is maintained. Antimicrobial treatment is usually continued until normalization of the C-reactive protein (generally up to 4–6 weeks in most cases), but, if the local signs of infection persist or re-appear, antimicrobials are stopped and the patient is referred again to the orthopedic surgeon.

Prosthetic joint infections were defined according to the EBJIS criteria for the “infection likely” or “infection confirmed” category [41]. Fracture-related infections (FRIs) were defined as the isolation of phenotypically indistinguishable microorganisms from the culture of at least 2 separate deep tissue/implant specimens in the presence of a sinus tract breakdown coupled with the presence of >5 polymorphonuclear cells per high-power field during the histopathology of intra-operative specimens [42]. Osteomyelitis was defined as the presence of an inflammatory process in the the bone caused by an infecting microorganism in the absence of any prosthetic material [1]. Septic arthritis was defined as the presence of a combination of clinical findings, physical examination and laboratory investigations of blood and synovial fluid [2,43].

Patients retrospectively included in this study received early empirical combination therapy based only on daptomycin (8–10 mg/kg daily) plus CI fosfomycin and were clinically assessed for our aims until switching from antimicrobial treatment to target or step-down therapy (namely more or less the first 2 weeks of treatment). Patients receiving combo therapy with agents other than daptomycin and those undergoing hemodialysis were excluded.

The preferred posology of CI fosfomycin changed over time in agreement with the findings of the previously mentioned population pharmacokinetic study [17]. Specifically, a standard 16 g daily CI fosfomycin dose was initially used. Reduced doses were applied after publishing a population pharmacokinetic/pharmacodynamics study showing that a dose of 8–12 g daily had likelihood >90% of attaining 70%t > MIC against *S. aureus* and CoNS across eGFR classes of 60–>130 mL/min/1.73 m^2^. These reduced doses were preferred at the discretion of the prescribing ID specialist.

After an initial 8 g loading dose over 1 h, patients having treatment before May 2023 received 16 g daily CI fosfomycin (high-dose group), whereas those having treatment thereafter received 8–12 g daily (reduced-dose group).

After 48–72 h from starting therapy, each patient underwent TDM for measuring fosfomycin plasma steady-state concentrations. Fosfomycin Css was targeted to >88.8 mg/L in order to achieve both 70%t > MIC [11,12,13] and AUC/MIC > 66.6 [15] against MRSA/MRSE with an MIC up to 32 mg/L, namely a value covering the vast majority of the MIC distribution of fosfomycin against staphylococci [44].

Plasma fosfomycin concentrations were measured by means of a liquid chromatography–tandem mass spectrometry (LS/MS-MS) analytic method, as previously described [45]. The intra- and inter-day coefficient of variations (CV) of the quality controls were 0.09–0.14% and 0.08–0.11%, respectively. The lower limit of quantification was 0.5 mg/L.

In each patient, fosfomycin AUC and clearance were calculated at each TDM assessment by means of the following Equations (1) and (2):(1)$$AUC\;\left(mg\times \frac{h}{L}\right)=Css\left(\frac{mg}{L}\right)\times 24\;h$$(2)$$CL\;(L/h)=\frac{IR\left(\frac{mg}{h}\right)\times 24\;h}{AUC\;(mg\times h/L)}$$
where AUC is the fosfomycin 24 h-area under-the-concentration versus the time curve (mg × h/L), CL is the fosfomycin clearance (L/h) and IR is the fosfomycin infusion rate (mg/h).

In patients undergoing multiple TDM assessments, the average clearance was calculated.

### 4.2. Data Collection

Demographic (age, gender, weight, height), clinical (type and site of infection, microbiological isolates, if available) and pharmacological data (fosfomycin dose, treatment duration and TDM values) were retrieved from electronical medical records. Laboratory parameters such as serum albumin, alanine aminotransferase, aspartate aminotransferase, sodium and potassium concentrations were collected at baseline and at the end of empirical treatment. Serum creatinine was assessed during each TDM session. The estimated glomerular filtration rate was calculated by means of the CKD-EPI formula [46].

### 4.3. Clinical Evaluation and Safety

Patient clinical assessments during antimicrobial treatment included a full blood chemistry examination with inflammatory bio-markers twice-weekly. Clinical efficacy was assessed at time of switching antimicrobial treatment and was considered as positive whenever local (rubor, tumor, calor, dolor) and/or systemic (fever and/or pain) signs of infection were improving vs. baseline and the C-RP was decreasing or normalizing in the absence of findings suggestive of worsening of the infection in imaging studies. Treatment inefficacy was defined when at least one of the previous criteria was not achieved [47].

### 4.4. Statistical Analysis

The Kolmogorov–Smirnov test was used to assess whether data were normally or non-normally distributed. Accordingly, means ± SD or medians with IQR were used in descriptive statistics. The statistical difference between groups was assessed by means of a chi-squared test or Fisher’s exact test, when required. Classification and a regression tree (CART) analysis was used to predict a cut-off value of fosfomycin AUC that best correlated with the probability of hypokalemia or hypernatremia occurrence. All statistical analysis and graphs were performed with R version 4.3.3 (The R Foundation for Statistical Computing, Vienna, Austria).

## 5. Conclusions

In conclusion, this study showed that, in patients with osteoarticular infections, 8 g daily of continuous infusion fosfomycin may reach optimal PK/PD target attainment with better safety than the use of the recommended 16 g daily dose in patients with preserved renal functions. Targeting fosfomycin Css between 88.8 and 347 mg/L was suggested for optimizing drug pharmacodynamics up to an MIC of 32 mg/L and minimizing the risk of hypernatremia and hypokalemia. Considering the high inter-individual variability of fosfomycin, TDM may represent a useful tool for optimizing drug exposure and reduce the risk of drug-related adverse events.

## Figures and Tables

**Figure 1 antibiotics-14-00139-f001:**
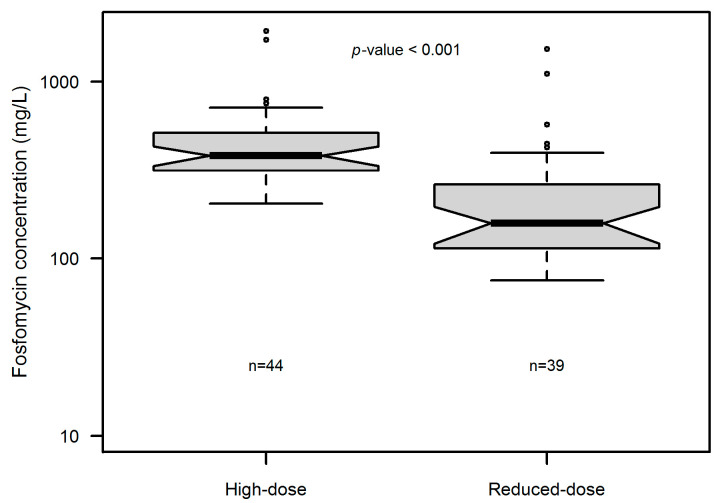
Box and whisker plot of the fosfomycin plasma concentration in patients treated with continuous infusion fosfomycin in the high-dose (*n* = 44) vs. reduced-dose (*n* = 39) regimen.

**Figure 2 antibiotics-14-00139-f002:**
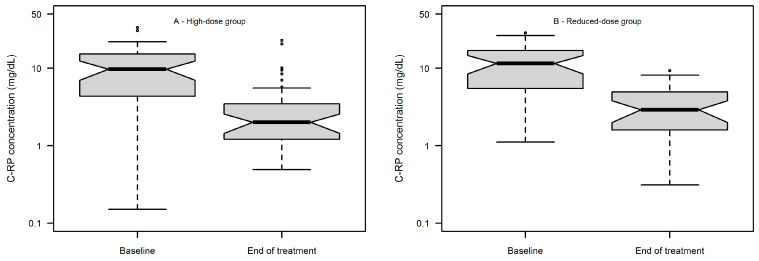
Box and whiskers plot of C-RP concentrations at baseline vs. at end of treatment in the high-dose group (*n* = 44, panel (**A**)) and in the reduced-dose group (*n* = 39, panel (**B**)).

**Figure 3 antibiotics-14-00139-f003:**
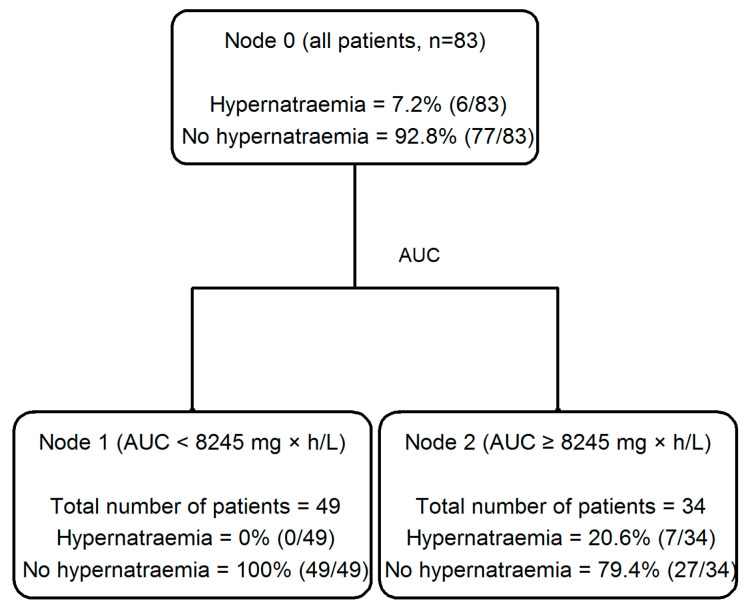
The classification and regression tree (CART) analysis of the partition of the occurrence of hypernatremia depending on fosfomycin AUC in patients with osteoarticular infections (*n* = 83).

**Figure 4 antibiotics-14-00139-f004:**
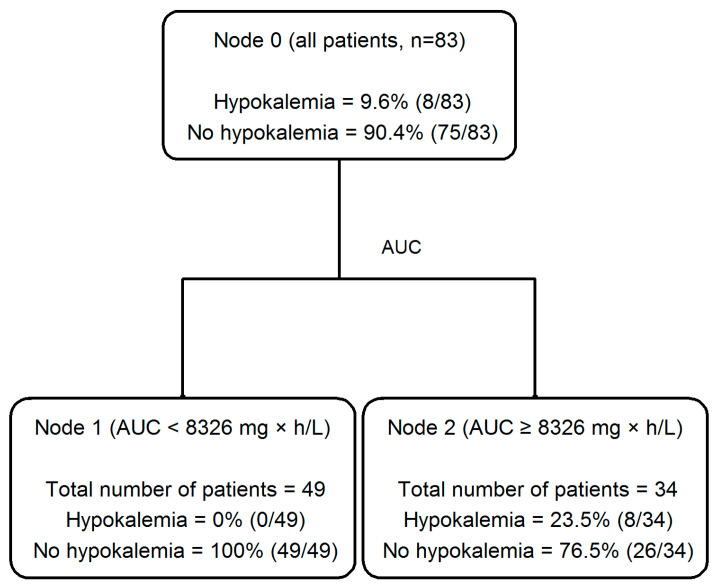
The classification and regression tree (CART) analysis of the partition of the occurrence of hypokalemia depending on fosfomycin AUC in patients with osteoarticular infections (*n* = 83).

**Table 1 antibiotics-14-00139-t001:** Demographic and clinical characteristics.

Variable		High-Dose (*n* = 44)	Reduced-Dose (*n* = 39)	*p*-Value
Age (years)		56 (46.5–67.5)	59 (49–65.75)	0.535
Gender (M/F)		30/14	23/16	0.369
Weight (kg)		80 (70.5–94)	76 (67.25–85)	0.178
BMI (kg/m^2^)		27.6 (24.4–31.2)	26.4 (22.9–30.5)	0.352
Albumin (g/dL)		3.52 (2.98–3.89)	3.23 (2.82–3.51)	0.137
AST (IU/L)		19 (15–25.5)	21(15.8–31)	0.421
ALT (IU/L)		17.5 (13–30)	18 (12.8–30.8)	0.907
Serum creatinine (mg/dL)		0.72 (0.60–0.81)	0.725 (0.65–0.93)	0.111
eGFR (mL/min/1.73 m^2^)		108 (101.25–115.75)	98.5 (90–109)	0.002
Diagnosis				
	Prosthetic joint infections	21 (47.7)	23 (58.8)	0.379
	Osteomyelitis	8 (18.3)	4 (10.3)	0.362
	Post-operative spinal implant infection	5 (11.4)	3 (7.7)	0.718
	Infected non-union	3 (6.8)	4 (10.3)	0.701
	Septic arthritis	3 (6.8)	3 (7.7)	1.000
	Vertebral osteomyelitis	2 (4.5)	1 (2.6)	1.000
	Skin and soft tissue infection	2 (4.5)	1 (2.6)	1.000
Microbiological isolates				
	MSSA	8 (18.2)	4 (10.3)	0.362
	MRSA	5 (11.4)	0 (0)	0.057
	CoNS	7 (15.9)	9 (23.1)	
	No isolates	22 (50)	26 (66.7)	0.181
Pharmacological treatment				
	Fosfomycin dose (g/daily)	16 (16–16)	8 (8–12)	<0.001
	Treatment duration (days)	8 (6–14)	11 (8–13)	0.220
	Steady-state concentration (mg/L)	381.7 (312.9–512.5)	157.8 (113.8–261.6)	<0.001
	Total fosfomycin clearance (L/h)	1.8 (1.3–2.2)	2.3 (1.4–3.1)	0.008
	Daptomycin dose (mg/daily)	700 (700–850)	700 (500–700)	<0.001
Clinical outcome				
	Clinical efficacy	40 (90.9)	35 (89.7)	1.000

ALT, alanine aminotransferase; AST, aspartate aminotransferase; BMI, body mass index; CoNS, coagulase-negative staphylococci; eGFR, estimated glomerular filtration rate; MSSA, methicillin-susceptible *Staphylococcus aureus*; MRSA, methicillin-resistant *Staphylococcus aureus*; TDM, therapeutic drug monitoring.

**Table 2 antibiotics-14-00139-t002:** Attainment of the pharmacokinetic/pharmacodynamic indexes of efficacy of 70%t > MIC and of AUC/MIC > 66.6 in relation to the MIC_50_ and MIC_90_ derived from the EUCAST distribution and also in relation to an MIC of 8 and 32 mg/L in patients who received high- vs. reduced-dose continuous infusion fosfomycin.

PK/PD Parameter		High Dose (*n* = 44)	Reduced Dose (*n* = 39)	*p*-Value
Fosfomycin MIC of 4 mg/L (MIC_50_)				
	70%t > MIC	44 (100)	39 (100)	1.000
	AUC/MIC > 66.6	44 (100)	39 (100)	1.000
Fosfomycin MIC of 8 mg/L				
	70%t > MIC	44 (100)	39 (100)	1.000
	AUC/MIC > 66.6	44 (100)	39 (100)	1.000
Fosfomycin of 16 mg/L (MIC_90_)				
	70%t > MIC	44 (100)	39 (100)	1.000
	AUC/MIC > 66.6	44 (100)	39 (100)	1.000
Fosfomycin MIC of 32 mg/L				
	70%t > MIC	44 (100)	39 (100)	1.000
	AUC/MIC > 66.6	44 (100)	36 (92.3)	0.099

AUC, area under-the-curve; eGFR, estimated glomerular filtration rate; MIC, minimum inhibitory concentration (mg/L); TDM, therapeutic drug monitoring.

**Table 3 antibiotics-14-00139-t003:** Type of adverse drug reactions observed in patients who received high- vs. reduced-dose continuous infusion fosfomycin.

Type of ADR	High Dose (*n* = 44)	Reduced Dose (*n* = 39)	*p*-Value
Hypernatraemia	5 (11.4)	1 (2.6)	0.207
Hypokalemia	7 (15.9)	1 (2.6)	0.061
Increased CPK	1 (2.3)	1 (2.6)	1.000
Gastrointestinal disorders	0 (0)	1 (2.6)	0.469
Hypertransaminasemia	0 (0)	2 (5.1)	0.217
Total adverse drug events	13 (29.5)	6 (15.4)	0.190

CPK, creatine phophokinase.

## Data Availability

The data presented in this study are available upon request from the corresponding author. The data are not publicly available due to privacy concerns. The de-identified individual participant data that underlie the results reported in this article (including text, tables, and figures) will be made available together with the research protocol for non-commercial, academic purposes. Additional supporting documents may be available on request.

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
