# Peer review of "Could a Reduced Dose of 8 g of Continuous Infusion Fosfomycin Be Considered as Effective as and Safer than a Standard 16 g Dose When Combined with High-Dose Daptomycin in the Treatment of Staphylococcal osteoarticular Infections?"

_antibiotics, 2025, doi:10.3390/antibiotics14020139_

Round 1
Reviewer 1 Report
Comments and Suggestions for Authors
Comments:
This study compared the probability of PK/PD target attainment and the occurrence of adverse events in patients receiving initial empirical treatment of osteoarticular infections with standard vs. reduced doses of continuous infusion fosfomycin combined with daptomycin. This study showed that a reduced dose may reach optimal PK/PD target attainment with better safety than the recommended 16g daily dose in patients with preserved renal function. This study is well organized, but some information still needs to be provided.
A few comments:
1. The sampling strategy and method to calculate the CL and AUC were not clearly described. Please add this information.
2. Please provide the r2 between fosfomycin clearance and eGFR.
3. In Table 1, the number of patients in the reduced dose group is 39, with 23 males and 17 females. The numbers cannot match.
Reviewer 2 Report
Comments and Suggestions for Authors
This is a relevant and innovative study that explores a promising alternative for treating osteoarticular infections. It also presents the judicious use of TDM (Therapeutic Drug Monitoring) as a control tool.
Here are some suggestions for improvement or adjustments required clarity in Study Objectives (Page 1, Lines 20-25):
-objectives should be made more direct, clearly highlighting the main research questions. Long sentences may make them difficult to understand.
Methodology (Page 9, Lines 314-354):
-Detail the selection of patients in the retrospective study in greater detail. How were ineligible cases excluded?
-Justify the reason for adopting specific doses of fosfomycin. Although there is a brief explanation, greater depth would add value.
Statistics (Page 11, Lines 386-391):
-The statistical analysis, although clear, could include the use of complementary tests or more robust analyses to reinforce the findings.
Discussion (Page 9, Lines 202-235):
-Consider additional limitations, such as bias related to the retrospective sample or possible influences of concomitant treatments.
-Expand the comparison with previous studies to better situate the results.
Conclusions (Page 14, Lines 394-401):
-Avoid extrapolations without concrete data. Phrases such as “may be appropriate” could be reinforced with more objective caveats.
Round 2
Reviewer 2 Report
Comments and Suggestions for Authors
no comments